# The Ecology-Economy-Transport Nexus: Evidence from Fujian Province, China

Wulin Wang [1], Jiao Gong [1], Wenyue Yang [2,*] and Jingyu Zeng [3]

1. College of Environment and Safety Engineering, Fuzhou University, 2 North Wulongjiang Avenue, Fuzhou 350108, China; wangwulin@fzu.edu.cn (W.W.); n190620009@fzu.edu.cn (J.G.)
2. College of Forestry and Landscape Architecture, South China Agricultural University, 483 Wushan Road, Guangzhou 510642, China
3. State Key Laboratory of Earth Surface Processes and Resource Ecology, Faculty of Geographical Science, Beijing Normal University, 19 Xinjiekou Wai Dajie, Beijing 100875, China; 202131051010@mail.bnu.edu.cn
* Correspondence: yangwenyue900780@163.com

**Abstract:** The coordinated relationship between ecology, economy and transportation is essential for regional sustainable development. Does the high-quality ecological environment mean the lagging development of economy and transportation, or does the rapid growth of the economy and transportation lead to the deterioration of the ecological environment? To shed new light on the complicated relationship between ecology, economy and transportation, our study aims to construct three comprehensive indicators, including an ecological index (EI), economic development level (EC) and transport superiority degree (TR), to reflect the systems mentioned above, and to measure the coordination of the three indicators' development and evolution using a model of the coordination degree (CD). Specifically, and by applying methods for the indicators' normalization, including superposition analysis and principal component analysis, the three indicators' values are reasonably evaluated for measuring their coordination relationship. The above three indicators use data from 58 counties in Fujian province from 2000 to 2018 in our study. All three indicators show differences in the west and east of Fujian province; the EI is relatively low in the eastern coastal areas and relatively high in the western mountainous areas, the EC shows a relatively discrete and irregular distribution and the distribution pattern of the TR is almost the opposite of the EI. The CD shows a relationship among the three indicators, with the EI and EC coordinated in most counties and the EI and TR coordinated in most counties, while the highly coordinated counties are mainly distributed in the northwest and east coastal regions of Fujian province in 2000, and the northwest, south and northeast of Fujian province in 2018. More than 50% of the county EC and TR values are kept in a coordinated state, and are mainly distributed in the eastern coast and central part of Fujian province. Over 50% of counties' CD between EI and EC, EI and TR and EC and TR are in a coordinated state. The CD of the EI and EC and TR, in most counties, are in a coordinated state, mainly distributed in the eastern coast and central areas of Fujian province. In other words, the findings show that the coordinated state of ecology, economy and transportation can be achieved at the county level of Fujian province. These conclusions have significant reference value for understanding regional sustainable development.

**Keywords:** ecological index; economic development level; transport superiority degree; coordination degree; county; Fujian province

## 1. Introduction

Globally, the sustainable development of regional ecological environments and the social economy has been the objective within all sectors (ecological, economic and socio-cultural) for continued human wellbeing [1]. The coordinated development of the ecological environment and the social economy is the common responsibility and pursuit of the international community [2]. Rapid economic growth and ecological degradation have increased

the focus on the sustainable relationship between ecological security and economic development [3]. In this study, the relationship between interrelated aspects of life, including the economy and ecology, environmental capacity concepts [4,5], ecological viewpoints [6] and sustainable development ideas [7], are widely applied. By establishing a comprehensive evaluation index system of economic development and the ecological environment, the economic development level and ecological environment quality can be comprehensively explored. Researchers have published varied interpretations of the ecological, economic, social relationship and outputs, including quantitative measurement of ecological service value and its spatial evolution [8–10], the economy and ecological environments [11], the verification of an environmental Kuznets curve [12] and society, ecology and tourism [13], and the ecological function, production function and living function [14]. The general understanding is that ecological environment degradation and economic backwardness are interrelated and geographically coupled, and that this coupling is presented in the form of a "poverty trap". More explicitly, the more serious the ecological environment degradation, the more backward the regional economy is, and vice versa, as economic backwardness further aggravates ecological environment degradation [15]. However, economic growth will also cause environmental pollution. For example, China's economic growth also leads to environmental pollution [16]. The relationship between economic growth and ecological environment improvement cannot be generalized. In different cities or regions, there is not only a harmonious relationship between an advanced economy and a good ecological environment, but also a low-level coupling and coordinated state, and the coordination between economy and environmental ecology is the necessary condition for sustainable development [17,18]. The high production and living functions, and the low ecology function mainly occurred in well-developed regions, as well as the substantial trade-offs between the above two functions and the ecology function [19]. Therefore, when making planning decisions for regional economic development, the concept of ecological relationships should be integrated with other factors to effectively protect and realize healthy and sustainable development outputs [20]. Huan et al. adapt the goal of sustainable development to include various dimensions of economy and ecological environment [21]. The coordinated development of the social economy and the ecological environment is a necessary condition for the sustainable development of urban areas [17]. However, on the whole, there are few studies on the relationship between economic development with ecological environment considerations at home and abroad, and we especially lack macro studies on the stages and pathways of regional environmental economic development [2].

Ecologists are increasingly attempting to resolve environmental impacts with considerations for significant infrastructure, large-scale experiments, observation networks and complex data and calculations [22]. A few studies investigated the rapid development of transportation infrastructure and economy, which impacted the ecological environment and contributed to greenhouse effects [23]. For example, the construction of railways and highways destroyed the natural vegetation and landscapes [24], causing various degrees of soil pollution along longitudinal lines [25]. In the long term, desertification may contribute substantial ecological risks to areas along longitudinal lines, such as rail tracks [26]. Dong et al. established an integrated risk evaluation model to analyze the economic, social and ecological risks of the China–Mongolia–Russia high-speed railway, determine their magnitude and spatial distribution pattern and propose policy suggestions to reduce construction risks [27]. Energy consumption related to the transportation sector is an integral part of all energy consumption, and the development of road infrastructure directly leads to an increase in carbon emissions in the transportation sector [28]. Related research regards transportation infrastructure as an essential indicator of resources and environmental carrying capacity, and then evaluates its roles in sustainable development [29]. Public infrastructure, including transportation infrastructure, is of great significance to protecting the ecological environment [30]. Some cities construct large-scale transportation infrastructure, but the ecological and environmental problems are more serious, the infrastructure

in these cities is not functioning as it should and it is not clear whether the environmental benefits of the infrastructure are coordinated [31].

The relationship between economy and transport has been at the frontier of economic and policy research [32]. Combining the development of transport infrastructure with economic growth, society and public institutions has the potential to produce great synergy [33]. The trend in all countries is to increase investment in transport infrastructure to stimulate economic growth and promote trade [34,35]. The practice has proved that high-level infrastructure investment is a harbinger of economic growth. This is especially exemplified in China [36], where transport and logistics infrastructure and economic development are in a long-term equilibrium state [37]. This combination has a driving effect on regional economic growth. However, different transport infrastructures, including rail and road, have different influences on regional economic development [38]. Although empirical studies in the countries along the "One Belt And One Road project" [39], emerging Asian economies [40], the Middle East and North Africa [41] and Spain [42] have shown a positive correlation between transport infrastructure and economic growth, it is still difficult to accurately summarize the potential impact of transport infrastructure on economic growth due to regional differences [43], while macroeconomic models are also different, and so it is necessary to conduct a general equilibrium analysis of transportation–economic links [44].

The role of ecology, economy and transportation has been highlighted in several academic research approaches, but few studies have evaluated the coupling and coordination degree of economy, the ecological environment and transportation in China from a regional perspective. Few studies have focused on the relationship between socio-economic development ecological and environmental quality, and there is a lack of macro-level research on the stage and path of regional environmental economic development [2]. There is an opportunity to further discuss the coupling and coordination relationship between the ecological environment and transport infrastructures. In particular, there is a lack of research on the coupling and coordination between the ecological environment, economic development and transport development levels. Given all this, by borrowing the concept of coordination degree and taking the counties of Fujian province as the basic unit, this study comprehensively studies the coordination relationship between the ecological environment, economic development level and transport superiority degree in a dynamic evolution from 2000 to 2018, and provides a theoretical basis for regional sustainable development. The remainder of this paper is organized as follows. Section 2 outlines the materials and methods used in the analysis. Section 3 discusses the results. Section 4 offers the concluding remarks of the study.

## 2. Materials and Methods

### 2.1. Study Area

Fujian province was established as the first ecological civilization pilot demonstration zone by the State Council in 2014. The province was elevated to the national strategic level, and this marked the entry of Fujian province into the system innovation stage of ecological civilization construction and regional ecological practice. Fujian province is located on the southeast coast of China, with a total land area of 121,400 square kilometers. The terrain is high in the northwest and low in southeast, and the province is surrounded by mountains and sea. Mountains and hilly areas account for 90% of the total territory. The four major river catchments include the Minjiang, Jinjiang, Jiulongjiang and Tingjiang rivers. The climate is a subtropical oceanic monsoon. Fujian province has 9 prefecture-level cities under its jurisdiction, including Fuzhou (provincial capital), Xiamen (special economic zone), Quanzhou, Zhangzhou, Putian, Longyan, Sanming, Nanping and Ningde, and has 12 county-level cities, 44 counties and 29 municipal districts (Figure 1), as of the end of 2020. The total resident population is 41.54 million, with a regional GDP of CNY 4.3903 trillion, ranking seventh among the provincial administrative units in China. The per capita GDP is CNY 110,500 which is ranked fourth in China.

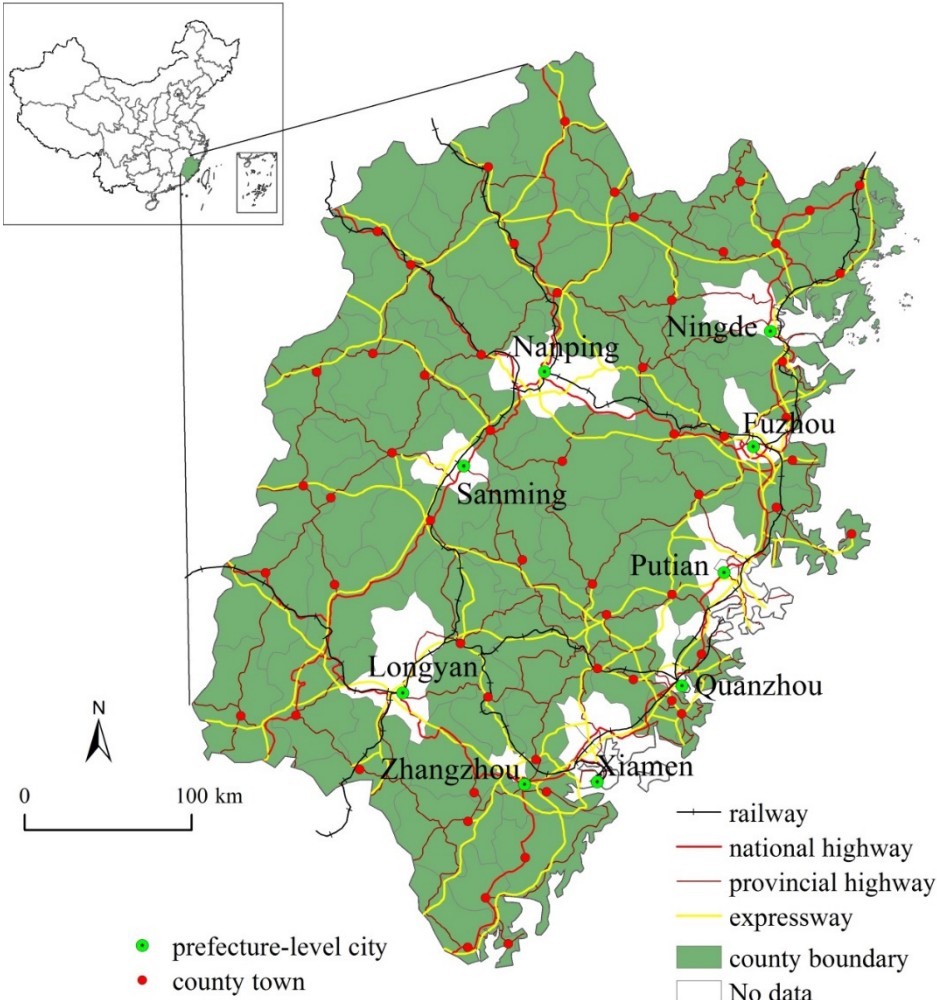

**Figure 1.** Location, cities, counties and land transportation infrastructure of Fujian province.

As the first ecological civilization pilot demonstration zone in China, Fujian province enjoys an ecological environment where the forest coverage was 66.8% in 2021. With rapid economic growth and continuous improvement of its infrastructure, the "most beautiful" ecological environment and the "high quality" of its economic development go hand in hand. Fujian province has developed into a province with strong provincial competitiveness in China, and has succeeded at building a society with well-balanced economic–environmental outcomes for human and ecological wellbeing. Fujian province is, with the implementation of its regional development policies, including being the core area of the 21st Century Maritime Silk Road, the Pilot Free Trade Zone, the Pilot Ecological Civilization Zone, the Fuzhou New Area, the Pingtan Comprehensive Experimental Zone, and the Fuxiaquan National Independent Innovation Demonstration Zone, expected to strongly support and promote high-quality developments. It should be pointed out that, since the fact that the districts attached to the 9 prefecture-level cities are the main built-up areas in Fujian province, and their natural background, economic development and infrastructure are pretty different from those of counties and county-level cities, the study area of this paper are the 58 counties and county-level cities in Fujian province, excluding Jinmen county and the districts attached to the 9 prefecture-level cities. In Figure 1, the "No data" areas mainly refer to the municipal districts of the 9 prefecture-level cities and Jinmen county.

*2.2. Data Sources*

This paper analyzes the ecological environment, economic and social development level and comprehensive transportation development level by using remote sensing data, statistical data and transport infrastructure network data from Fujian province. All remote sensing data reflecting the ecological environment quality in this study refer to land use data, NDVI, biological richness, vegetation coverage, water network denseness and land stress from MODIS and Landsat 8 (https://ladsweb.modaps.eosdis.nasa.gov/search/, accessed on 8 November 2019; https://www.nasa.gov/mission_pages/landsat/main/index.html, accessed on 8 November 2019). The NDVI data use the MOD13A1 product, with a spatial resolution of 500 m and a time resolution of 16 days, which are synthesized into annual values through the MVC method. The land use data use MCD12Q1 products, with a spatial resolution of 500 m, the data years are 2000 and 2018, and the land use classification standard is IGBP. The surface reflectance data use the MOD09A1 product with a spatial resolution of 500 m. The daily value product is mainly crowded to calculate the surface albedo, and then estimate the degree of land erosion. The data reflecting the economic and social development levels were drawn from the statistical yearbooks of Fujian province and the relevant prefecture-level cities in Fujian province from 2001 to 2019. The transport infrastructure network data were obtained by vectorizing the transport facilities networks according to the "Atlas of China by Provinces" (China Map Publishing House, 2001), and the "Series Atlas of Chinese Provinces: Atlas of Fujian Province" (Planet Map Publishing House, 2019).

*2.3. Methods*

2.3.1. Ecological Index

The ecological environment evaluation is a comprehensive index. Referring to the "Technical Criterion for Ecosystem Status Evaluation" of the Ministry of Environmental Protection of China [45], the ecological index (EI) is calculated using Formula (1). The overall state of the regional ecological environment is reflected in the EI. According to the availability and representativeness of the specific indicators, concerning the "Technical Criterion for Ecosystem Status Evaluation" issued by the Ministry of Environmental Protection of the People's Republic of China in 2015, and the sub-indices, including the biological richness index (BAI), vegetation coverage index (VCI), water network denseness index (WNI) and land stress index (LSI), the EI is calculated as follows after modification:

$$\mathrm{EI} = 0.35 \times \mathrm{BAI} + 0.30 \times \mathrm{VCI} + 0.20 \times \mathrm{WNI} + 0.15 \times (100 - \mathrm{LSI}) \tag{1}$$

$$\mathrm{BAI} = (\mathrm{BI} + \mathrm{HQ})/2 \tag{2}$$

$$\mathrm{HQ} = A_{\mathrm{bio}} \times (0.35 \times \text{forest land} + 0.21 \times \text{grassland} + 0.28 \times \text{wetland} + 0.11 \times \text{arable land} + 0.04 \times \text{construction land} + 0.01 \times \text{unused land})/\text{regional area} \tag{3}$$

$$\mathrm{VCI} = \text{NDVI of regional average} = \times \left( \frac{\sum_{i=1}^{n} P_i}{n} \right) \tag{4}$$

$$\mathrm{WNI} = (A_{\mathrm{riv}} \times \text{river length}/\text{regional area} + A_{\mathrm{lak}} \times \text{water area}/\text{regional area} + A_{\mathrm{res}} \times \text{water resources}/\text{regional area})/3 \tag{5}$$

$$\mathrm{LSI} = A_{\mathrm{ero}} \times (0.40 \times \text{severe erosion area} + 0.20 \times \text{moderate erosion area} + 0.20 \times \text{construction land area} + 0.20 \times \text{other stressed land area})/\text{regional area} \tag{6}$$

Full details, determinants and explanations for the above equations are provided below:

BAI is a biological richness index, evaluating the richness and poverty of living things in the region, comprehensively expressed by assessing the quality of living places and biodiversity. It consists of the biodiversity index (BI) and the habitat quality index (HQ). The BI is determined using the regional biodiversity evaluation standard, and the evaluation of BI includes the richness of wild vascular plants, the richness of wild animals, the diversity of ecosystem types, species specificity, the richness of threatened species and the invasion degree of exotic species [46]. In Formula (3), HQ is used to evaluate the suitability of the habitat quality of the main protected objects in a nature reserve, which is expressed

by the habitat quality of the main protected objects. $A_{bio}$ is the normalized coefficient of the habitat quality index, and the reference value is 511.26. In Formula (4), VCI is used to evaluate the degree of regional vegetation coverage, which is expressed based on the normalized vegetation index per unit area (NDVI). $P_i$ is the average value of the maximum NDVI of pixels from May to September, n is the number of regional pixels and $A_{veg}$ is the normalized coefficient of vegetation coverage index, with the reference value of 0.01. In Formula (5), WINI is used to evaluate the water richness in the region, which is expressed based on the total length of rivers, water area and water resources per unit area in the evaluation region. $A_{riv}$ is the normalized coefficient of river length and the reference value is 84.34, $A_{lak}$ is the normalized coefficient of water area and the reference value is 591.79 and $A_{res}$ is the normalized coefficient of water resources and the reference value is 86.39. In Formula (6), LSI is used to evaluate the degree of stress on the land quality in the region, which is expressed by the stress types, such as soil erosion, land desertification and land development per unit area, in the evaluation region. $A_{ero}$ is the normalized coefficient of the land stress index and the reference value is 236.04.

2.3.2. Economic Development Level

The method of measuring the level of county economic development was calculated by referring to relevant literature, and considering the generality, comparability and data availability of the indicators. This paper constructs the evaluation index system for the county economic development level (EC) using four dimensions and sixteen indicators [47] (Table 1).

**Table 1.** Dimension and indicator of economic development level for PCA.

| Dimension | Indicator | Unit |
|---|---|---|
| income and purchasing power | per capita GDP | CNY/person |
| | per capita savings deposit balance of residents | CNY/person |
| | per capita total retail sales of social consumer goods | CNY/person |
| | per capita net income of farmers | CNY |
| non-agricultural industry development | ratio of added value of tertiary industry to that of the secondary industry | - |
| | number of employees in the secondary industry | persons |
| | number of employees in the tertiary industry | persons |
| | number of industry enterprises above designated size per 10,000 people | 1/10,000 people |
| agricultural development | average total power of agricultural machinery | 1000 kw/100 km$^2$ |
| | per capita total agricultural output value | CNY 10,000/person |
| | per capita grain output | kg/person |
| government ability | per capita fixed assets investment | CNY 10,000/person |
| | per capita public finance income | CNY 10,000/person |
| | per capita public finance expenditure | CNY 10,000/person |
| | beds in medical and health institutions per 1000 people | beds/1000 persons |
| | welfare beds per 1000 people | beds/1000 persons |

Principal component analysis (PCA) is a statistical method that transforms several related numerical indicators into a few unrelated comprehensive indicators through the dimension reduction process, that is, using fewer indicators to replace and comprehensively reflect the original information, and these comprehensive indicators are the main components of the original multi-indicators. Using the principal component analysis (PCA) method, this paper calculates the economic development level scores of the counties in Fujian province in 2000 and 2018.

The main advantages of PCA are as follows: (1) it can eliminate the correlation between indicators; (2) if the number of original indicators is large, a few comprehensive indicators are selected to replace the original indicators on the premise of reserving the vast majority of information; (3) the weights determined by the PCA are objective and reasonable.

In particular, the main steps of PCA are as follows: (1) To judge whether factor analysis is necessary, the KMO test and Bartlett's spherical test are used to determine whether the analysis requirements are met. It is generally considered appropriate when the KMO is greater than 0.70, and the sig. of Bartlett's sphericity test is less than 0.05. (2) Calculate eigenvalues and eigenvectors, and extract principal components with eigenvalues greater than 1 and a variance contribution rate reaching 85% cumulatively. (3) Identify the contribution rate and cumulative contribution rate of principal components and calculate the total score composed of each principal component. All the above steps are completed in SPSS software.

### 2.3.3. Transport Superiority Degree

The transport superiority degree (TR) is an integrated index to evaluate the regional transport superiority, using three aspects with the same weighting: the scale of the regional transport infrastructure network, the influence degree of traffic trunk lines, and the accessibility status of the region within the overall macroscopic transport infrastructure network. In this study, the scale of the regional transport infrastructure network is measured by the density of expressways, national roads and provincial roads with equal weighting. The influence degree of the traffic trunk lines includes the distance relationship of expressways, railways, ports and airports (Table 2). The accessibility status of the region in the overall macroscopic transport infrastructure network means location dominance, which is measured by the minimum time cost to cities above the prefecture level. In general, the greater the value of the transport superiority degree, the more obvious its advantage.

**Table 2.** Indicator weight of influence degree of traffic trunk lines.

| Expressway | Weight | Railway | Weight | Port | Weight | Airport | Weight |
|---|---|---|---|---|---|---|---|
| own expressway | 3 | own railway | 3 | own hub port | 2 | own trunk airport | 2 |
| own national road | 2 | within 30 km from the railway | 2 | own general port | 1 | own regional airport | 1 |
| within 30 km from the expressway | 1.5 | within 60 km from the railway | 1 | within 30 km from hub port | 0.5 | within 30 km from trunk airport | 0.5 |
| within 60 km from the expressway | 1 | other | 0 | other | 0 | other | 0 |
| other | 0 | - | | - | | - | |

### 2.3.4. Coupling Coordination Degree Model

The concept of system coupling is derived from physics, and refers to the phenomenon that two or more systems interact to achieve synergy. According to Zhang and Wen [48], the coordination degree is one of the critical tools with which to study the coordinated development of the ecological environment and economy. Based on the pertinent literature (see references [17,20]), the calculation process of coupling degree is as follows:

$$C_n = \left[ \frac{U_1 \times U_2 \times \cdots \times U_n}{\prod_{i \neq j}(U_i + U_j)} \right]^{\frac{1}{n}} \tag{7}$$

In which the coupling degree value $C_n \in [0, 1]$. $U_1$, $U_2$ and $U_3$ can refer to EI, EC and TR, respectively. When two subsystems, such as EI and EC, are measured, the value of n is 2, and when three subsystems, such as EI, EC and TR, are measured, the value of n is 3. The deficiency of the coupling degree model is that it is difficult to reflect the development level of subsystems. For example, when the development level of subsystems is low, it can also be concluded that the coupling degree of subsystems is high. The coordination degree model can better estimate the coordination degree of interactive coupling among subsystems, compared to the coupling degree model. The coordination degree model is calculated as follows:

$$CD = (C_n \times T)^{\frac{1}{n}} \tag{8}$$

$$T = \alpha_1 \times U_1 + \alpha_2 \times U_2 + \alpha_3 \times U_3 + \ldots + \alpha_n \times U_n \tag{9}$$

where, CD is the coordination degree, T is the coordination index among subsystems and $\alpha_{1\sim n}$ is the undetermined coefficient. An assumption in this paper is that EI, EC and TR are equally important. When calculating the CD values of any two subsystems, $\alpha_1$ and $\alpha_2$ are $\frac{1}{2}$ (0.5), and when calculating the CD values of the three subsystems, $\alpha_1$, $\alpha_2$ and $\alpha_3$ are all 1/3 (0.34). Generally, CD can be categorized into six types [49,50]: (1) $0 < CD \leq 0.3$, extreme incoordination (EXI); (2) $0.3 < CD \leq 0.4$, intermediate incoordination (INI); (3) $0.4 < CD \leq 0.5$, basic incoordination (BAI); (4) $0.5 < CD \leq 0.6$, primary coordination (PRC); (5) $0.6 < CD \leq 0.7$, intermediate coordination (INC); (6) $0.7 < CD \leq 1$, quality coordination (QUC).

## 3. Results

### 3.1. Evolution Patterns of EI, EC and TR and Their Pairwise CD

3.1.1. The Pattern and Evolution of the Three Comprehensive Indicators

According to Formula (1), the EI values of counties in Fujian province in 2000 and 2018 are calculated. The KMO test values in 2000 and 2018 are 0.748 and 0.721, respectively, and the sig. of Bartlett's sphericity test is 0.000, which shows that the 16 indexes of EC are all suitable for factor analysis. In 2000 and 2018, the first four principal component eigenvalues are greater than 1, and the cumulative contribution rates reach 77.39% and 71.50%, respectively, and the factor scores of EC in 2000 and 2018 are obtained by multiplying the contribution rate with the principal component factor scores. The TR values of counties in Fujian province in 2000 and 2018 are measured by the scale of the regional transport infrastructure network, the influence degree of traffic trunk lines and the accessibility status of the region. After standardization, the EI, EC and TR of counties in Fujian province ranged from 0 to 1 (Figure 2). During the period from 2000 to 2018, the EI of counties in Fujian province generally maintained a relatively low pattern in the eastern coastal areas and a relatively high pattern in the western mountainous areas, while the EC values are relatively discrete and irregularly distributed on the plot/graphs and the EC in the border counties of the province was generally low. The advantage of the EC in the eastern coastal counties was obvious in 2018. The TR generally indicated an opposite distribution characteristic to the EI. Namely, it showed a relatively high pattern in the eastern coastal areas and a relatively low pattern in the western mountainous areas.

3.1.2. Evolution of CD between EI and EC

This measure aims to quantitatively reflect the coordinated development degree of the ecological environment and the economy, i.e., whether they are in a state of coordination or incoordination, and take timely adjustment and control measures according to the changing trend of the coordination degree to provide a theoretical basis for regional sustainable development. The changes in the coordination degree types between EI and EC in the counties of Fujian province from 2000 to 2018 are shown in Figure 3a. By the division standard of 0.5, the number of coordinated counties decreased from 53 in 2000 to 51 in 2018, and the number of uncoordinated counties increased from 5 to 7 accordingly. Specifically, QUC decreased from 26 to 12, INC increased from 20 to 24, PRC increased from 7 to 15,

BAI increased from 2 to 5, INI decreased from 2 to 1 and EXI remained unchanged with 1 in 2000 and 2018.

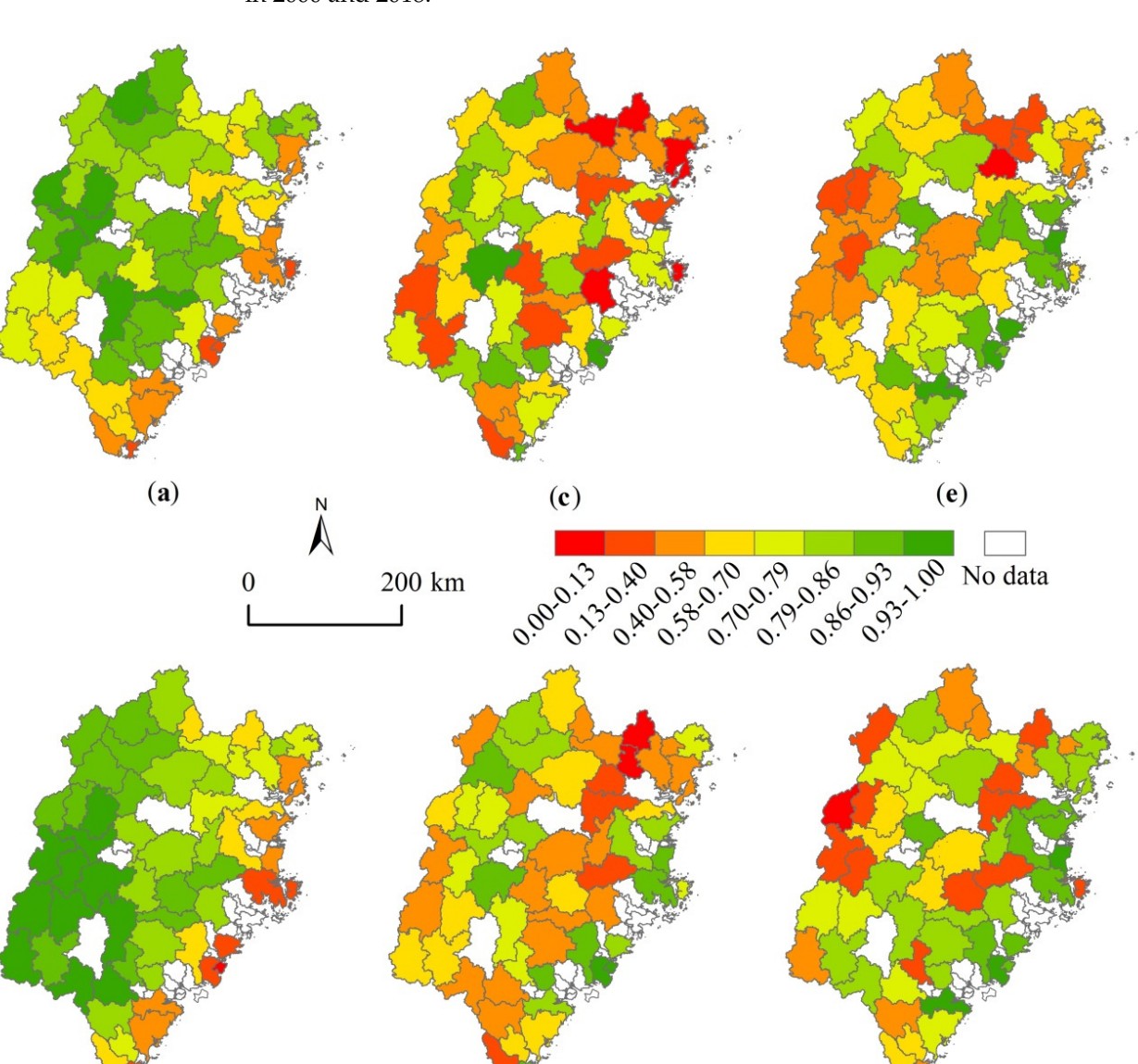

**Figure 2.** Evolution of EI and EC and TR in Fujian province in 2000 and 2018. (**a**) EI of 2000; (**b**) EI of 2018; (**c**) EC of 2000; (**d**) EC of 2018; (**e**) TR of 2000; (**f**) TR of 2018.

Regarding the spatial distribution of the coordination degree between EI and EC in Fujian province in 2000 and 2018 (Figure 4a,b), the counties with a CD above 0.5 are mainly located in the west and middle, while those with a CD below 0.5 are mainly located by the east coast and in the north.

### 3.1.3. Evolution of CD between EI and TR

Ecological and environmental problems, such as greenhouse effects from increased carbon emission, have increased with the rapid development of transport infrastructure initiatives [23]. The nexus of transport, ecological and environment interactions has become an important research topic to investigate relationships and causative contributing factors for seeking appropriate solutions. The changes to the coordination degree types between EI and TR in the counties of Fujian province from 2000 to 2018 are shown in Figure 3b. According to the division standard of 0.5, the number of coordinated counties increased

from 54 in 2000 to 56 in 2008, and the number of uncoordinated counties simultaneously decreased from 4 to 2. Specifically, QUC increased from 22 to 28, INC increased from 22 to 25, PRC decreased from 10 to 3, BAI decreased from 3 to 2, INI fell from 1 to 0 and EXI was 0 in 2000 and 2018.

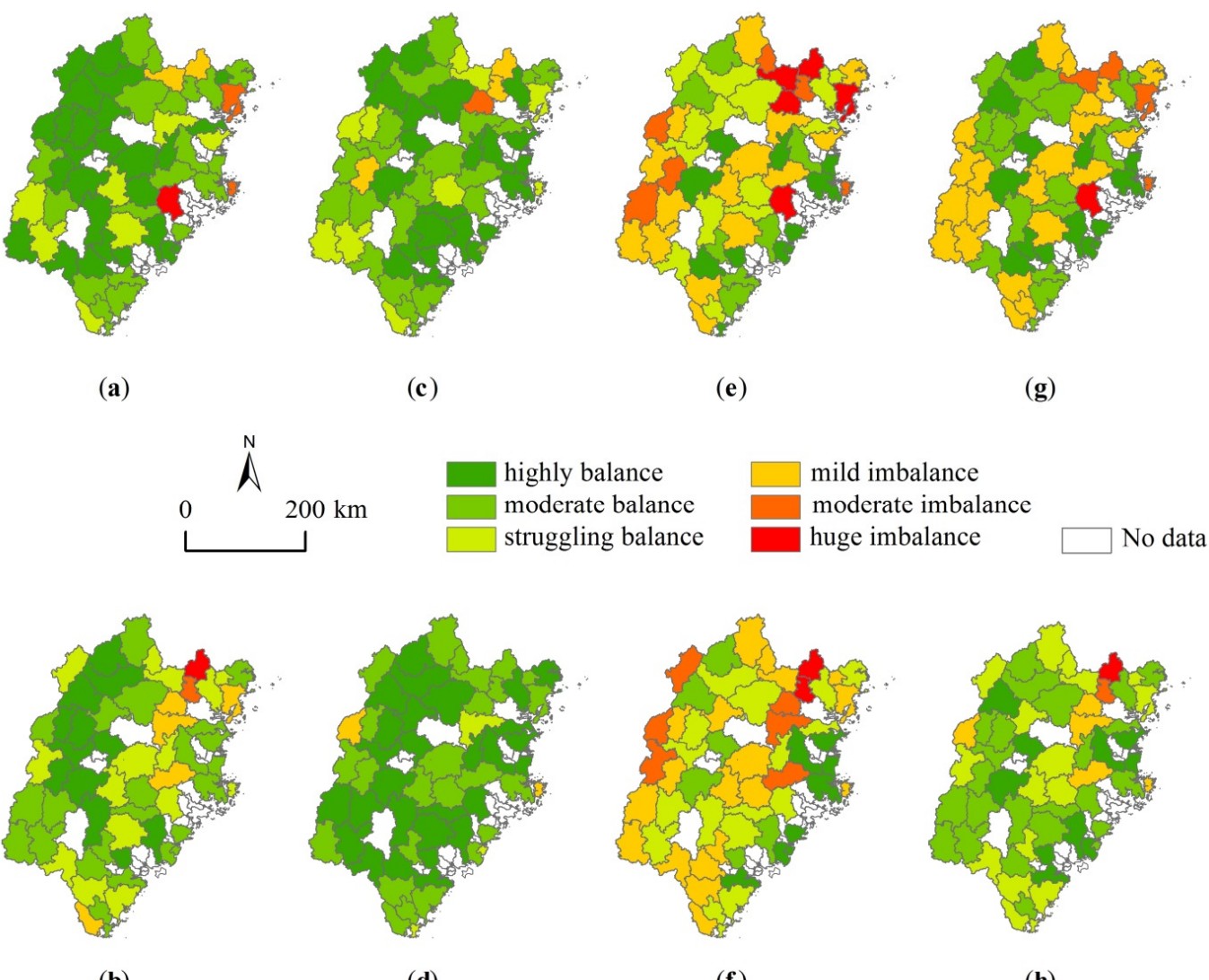

**Figure 3.** Coordination type of the 58 counties in Fujian province in 2000 and 2018. (**a**) Coordination type of EI & EC in 2000; (**b**) Coordination type of EI & EC in 2018; (**c**) Coordination type of EI & TR in 2000; (**d**) Coordination type of EI & TR in 2018; (**e**) Coordination type of EC & TR in 2000; (**f**) Coordination type of EC& TR in 2018; (**g**) Coordination type of EI & EC & TR in 2000; (**h**) Coordination type of EI & EC & TR in 2018.

According to the spatial distribution of CD between EI and TR in Fujian province from 2000 to 2018 (Figure 4c,d), most counties were in different degrees of coordination when the CD was above 0.5, and only a few counties were in different degrees of uncoordinated when the corresponding CD was below 0.5. The QUC counties were mainly distributed in the northwest and east coast in 2000, and mainly distributed in the northwest, south and northeast in 2018.

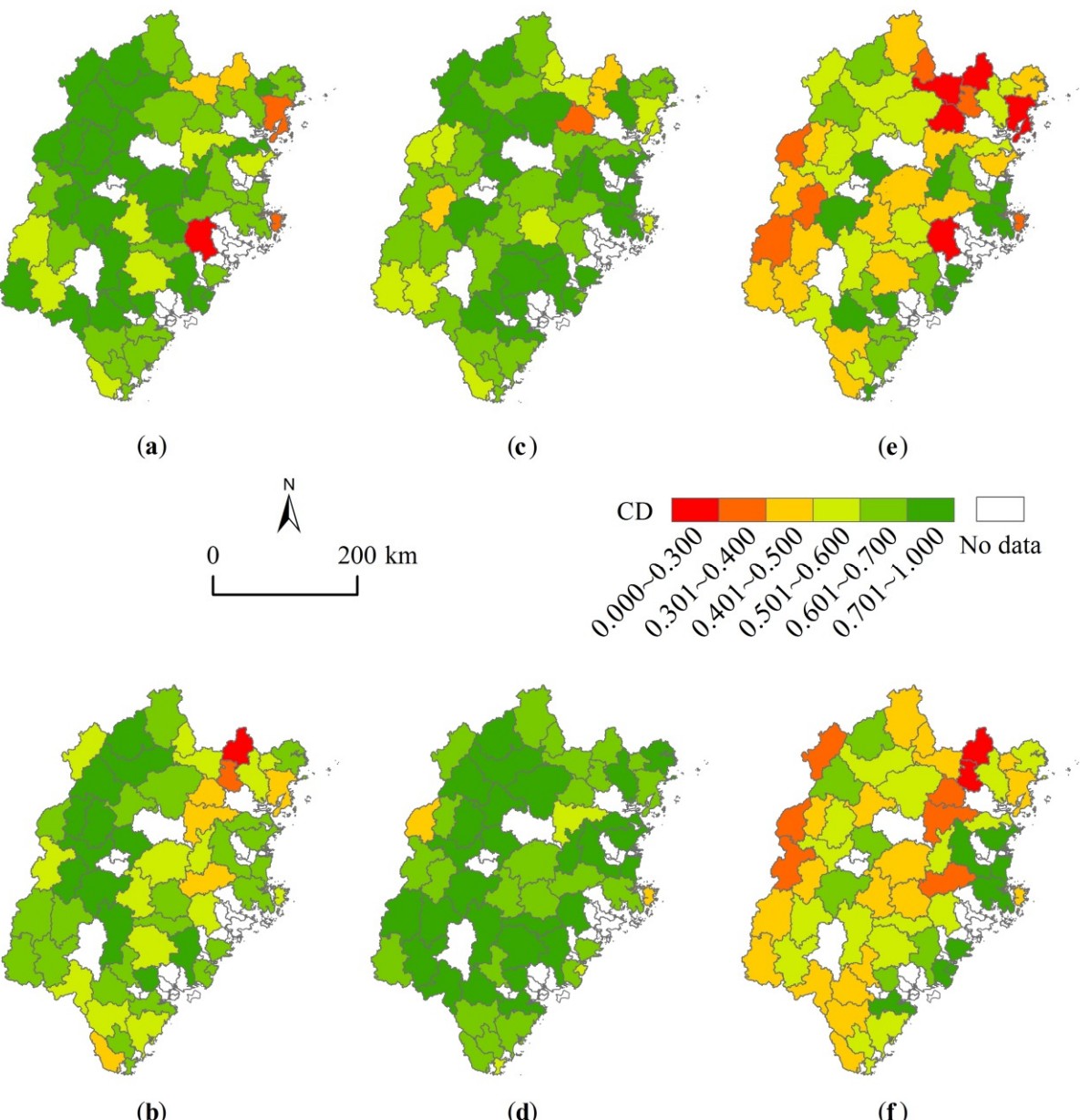

**Figure 4.** Coordination degree between EI and EC and TR in Fujian province in 2000 and 2018. (**a**) Co-ordination degree of EI & EC in 2000; (**b**) Coordination degree of EI & EC in 2018; (**c**) Coordination degree of EI & TR in 2000; (**d**) Coordination degree of EI & TR in 2018; (**e**) Coordination degree of EC & TR in 2000; (**f**) Coordination degree of EC & TR in 2018.

### 3.1.4. Evolution of CD between EC and TR

The development of transport infrastructure when combined with other policy inter-ventions has the potential for transformational impacts on the economy [51]. According to Lakshmanan [44], it is difficult for macroeconomic model evaluations to reveal the causal mechanism between transportation and the economy. The coordination degree model can accurately evaluate the degree of coordination between transport and the economy, addressing the uncertainties of causal mechanisms. The changes to the coordination degree types between EC and TR in the counties of Fujian province from 2000 to 2018 are shown in Figure 3c. Using the division standard of 0.5, the number of coordinated counties decreased from 32 in 2000 to 31 in 2018, and the number of uncoordinated counties increased from 26 in 2000 to 27 in 2018. Specifically, QUC decreased from 11 to 8, INC decreased from 7 to 6,

PRC increased from 14 to 17, BAI increased from 15 to 19, INI was 6 in 2000 and 2018 and EXI decreased from 5 to 2.

The spatial distribution of the CD between EC and TR from 2000 to 2018 is shown in Figure 4e,f. Counties with a CD between EC and TR above 0.5 have different degrees of coordination, and are mainly distributed by the eastern coast and central parts in 2000 and 2018, while the counties with a CD below 0.5, reflecting different degrees of incoordination, are mainly distributed in the northern and western areas for the same period.

### 3.2. Quantitative Change in CD between EI and EC, EI and TR and EC and TR

The changes of three groups of CD and ecological environmental indices were calculated, namely, CD and EI and *EC*, CD and EI and TR, CD and EC and TR, from 2000 to 2018. The Results are shown in Figure 5. All the three groupings of CD values above 0.5 with different degrees of coordination for the counties in 2000 were 32, which decreased to 30 in 2018. In 2000, the number of counties with a CD value above 0.5 of any two groupings among the three group combinations was 17, and this increased to 19 in 2018. The number of counties with a CD value above 0.5 of any one group among the three group combinations was seven in 2000, and this increased to nine in 2018. The number of counties with CD value below 0.5 of all the three groupings was two in 2000 and zero in 2018.

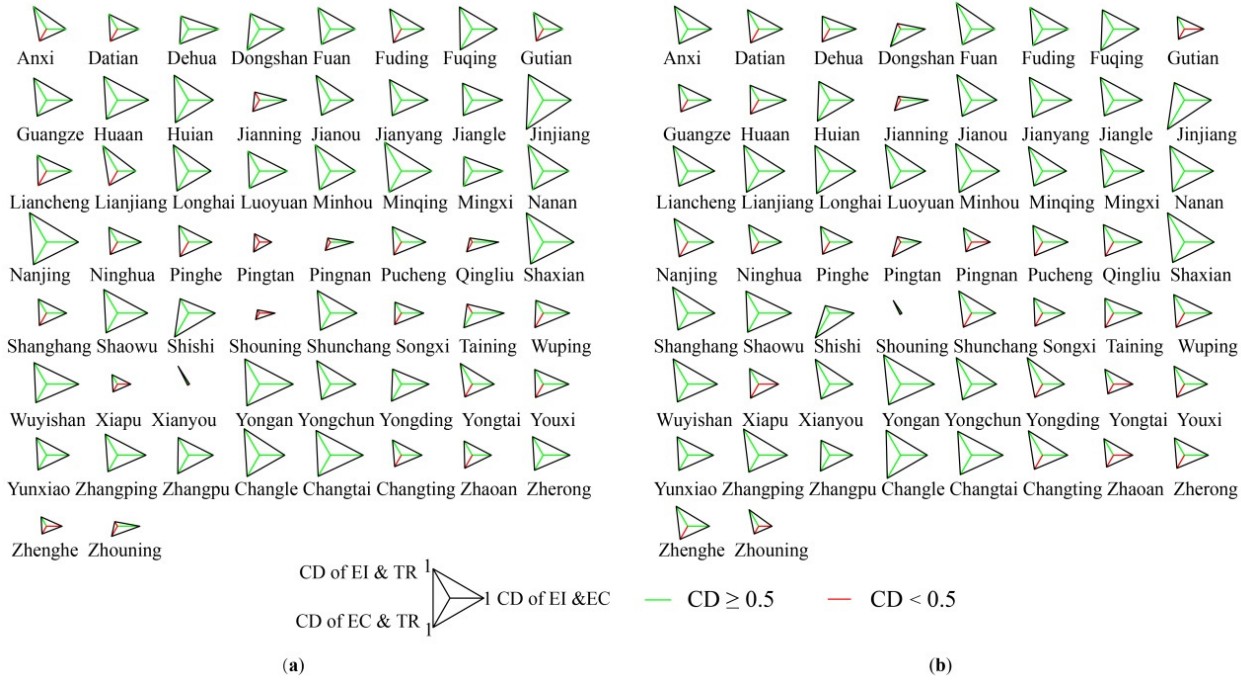

**Figure 5.** Coordination degree between EI, EC and TR at the county level in Fujian province in 2000 and 2018. (**a**) 2000; (**b**) 2018.

### 3.3. Evolution Pattern of Coupling Degree and CD and EI and EC and TR

There are three systems of ecological environment, economy and transport, which have mutual influence, mutual restriction and coexistence. Infrastructure, ecological environment and industrial economic development promote and restrict each other, and ignoring any links will affect regional sustainable development [52]. The calculation of the coupling degree is the premise of obtaining the CD. Using the radar charts of the coupling degree and the CD in 2000 and 2018, as shown in Figure 6, the coupling degree and the CD of the EI and EC and TR are almost in similar forms during the two years 2000 and 2018, but the value of the coupling degree is trending higher than that of the CD. In 2000 and 2018, the coupling degree of most counties ranged from 0.7 to 1, while the corresponding CD ranged from 0.5 to 0.7. Although the value of the coupling degree is greater than 0.5 with coupling coordination, the CD of counties may not be in a coordination state.

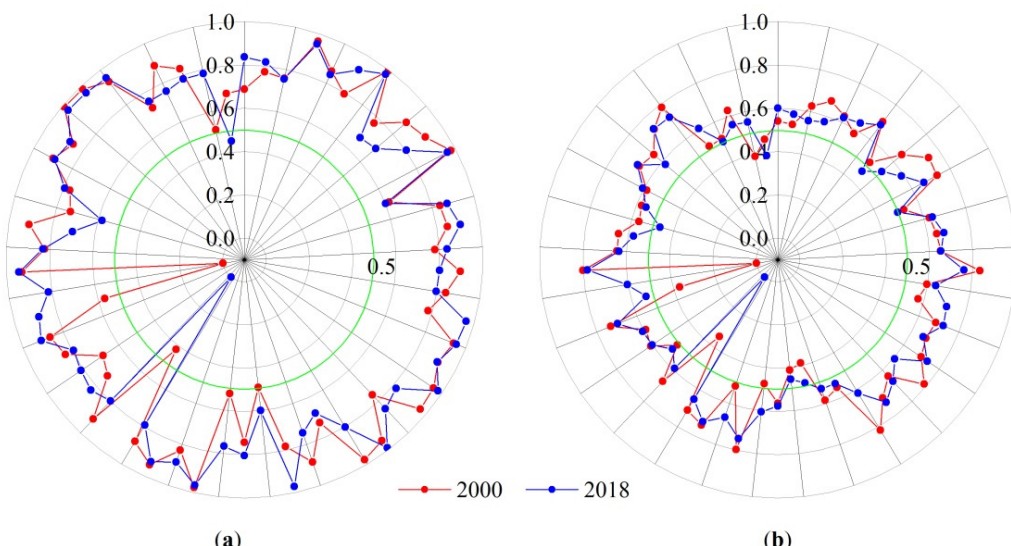

(**a**)                   (**b**)

**Figure 6.** Comparison between coupling and coordination degree of EI and EC and TR in Fujian province in 2000 and 2018. (**a**) Coupling degree of EI & EC &TR; (**b**) Coordination degree of EI & EC &TR.

The changes to the CD types among EI and EC and TR from 2000 to 2018 are shown in Figure 3d. Using the division standard of 0.5, the number of coordinated counties increased from 50 in 2000 to 51 in 2018. The number of uncoordinated counties decreased from eight to seven. For the other parameters, QUC decreased from 14 to 10, INC increased from 18 to 24, PRC decreased from 18 to 17, BAI increased from 3 to 5, INI decreased from 4 to 1 and EXI was 1 in the two years examined.

The spatial distribution of the CD among EI and EC and TR in 2000 to 2018 is shown in Figure 7, revealing that the counties with a CD above 0.5 in the two years examined are mainly distributed in the eastern coastal and central areas, while the counties with a CD below 0.5 are mainly distributed in the northern and western areas.

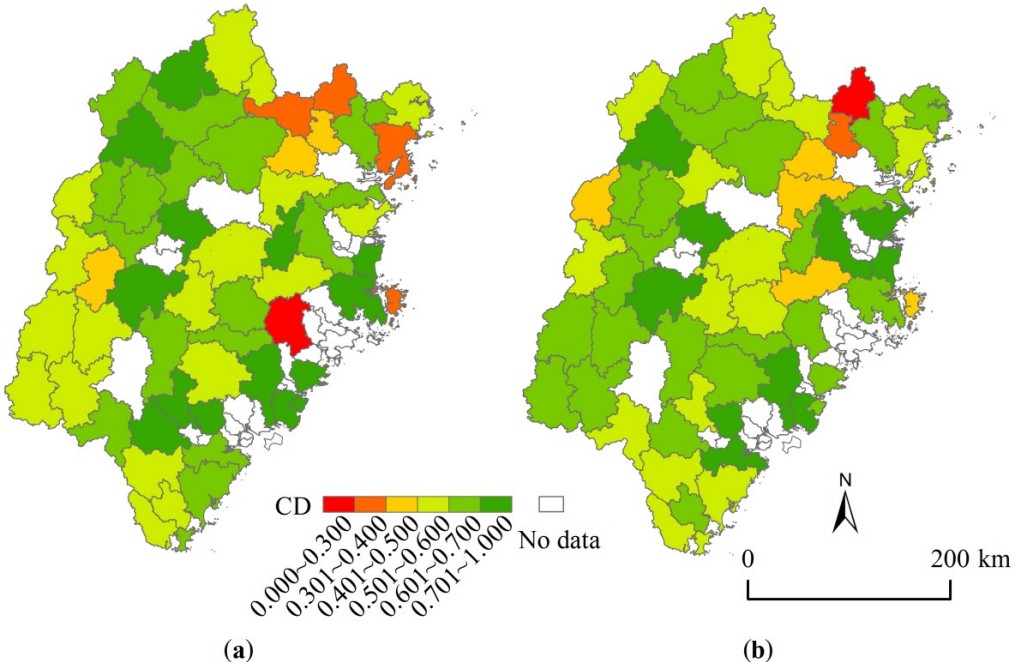

(**a**)                   (**b**)

**Figure 7.** Coordination degree of EI and EC and TR in Fujian province in 2000 and 2018. (**a**) Coordination degree of EI & EC & TR in 2000; (**b**) Coordination degree of EI & EC & TR in 2018.

## 4. Discussion and Conclusions

The primary purpose of this paper is to construct three integrated indicators of EI, EC and TR to measure the level and evolution of county ecological, economic and transportation development in Fujian province in 2000 and 2018, and to measure the CD evolution of the above three indicators. As the first ecological civilization pilot demonstration zone of China, it is extremely essential to carry out systematic research on the relationship between ecological quality, economic level and comprehensive transportation in Fujian province.

The results revealed that the EI maintains the pattern of being lower by the east coast and higher in the west mountain area. The distribution pattern of the EC is relatively disordered. The EC is usually lower in border counties and generally higher in eastern coastal counties. The distribution pattern of the TR is opposite to the EI; specifically, it is higher by the east coast and lower in the western mountain areas.

The results reveal that the CD between EI and EC is greater than 0.5 in most counties. The coordinated counties are mainly distributed in the western and central areas, and the uncoordinated counties are distributed mostly in the eastern coastal and northern areas. The CD between EI and TR is correspondingly greater than 0.5 in most counties. Counties with a CD between EI and TR in QUC states are mainly distributed in northeast and east coast areas in 2000, and south and northeast areas in 2018. The number of counties with a coordinated CD between EC and TR dropped from 32 in 2000 to 31 in 2018. The coordinated counties are mainly located in the eastern coastal and central areas, while uncoordinated counties are mainly distributed in the northern and western areas.

Regarding the three groups of CD combinations of EI and EC, EI and TR and EC and TR, the number of counties with a CD greater than 0.5 decreased from 32 in 2000 to 30 in 2018, the number of counties with a CD greater than 0.5 in any two groups increased from 17 in 2000 to 19 in 2018 and the number of counties with a CD greater than 0.5 in only one group increased from 7 to 9. The number of counties with a CD of the three groups all lower than 0.5 decreased from two in 2000 to zero in 2018.

The CD between EI and EC and TR in most counties is in a coordinated state, given that the number of counties with a CD greater than 0.5 increased from 50 in 2000 to 51 in 2018. The counties in a coordinated state are mainly distributed in the eastern coastal and central areas. In contrast, the counties in an uncoordinated state are mainly distributed in the northern and western areas.

### 4.1. Contributions

This paper makes several significant contributions to the literature. First, the parameters used and the calculation method developed in this study provide sufficient indicators to track the sustainability of the ecological environment, economic development and transport development, and reveal the dynamic coordinated relationship among the three. This study proves that ecology, economy and transport can co-exist and develop harmoniously.

Second, the existing literature includes research into the coupling and coordination relationship between ecology and economy, ecology and transportation and economy and transportation. Still, few studies pays attention to the coupling and coordination relationship among the three complex systems of ecology, economy and transportation. This paper supplements the research with content concerning the complex relationship between these systems.

Third, although our research paradigm has not received enough attention, it is valuable for straightening out the relationship between county ecology, economy and transportation, and formulating policies for ecological environment governance, economic and transportation development.

### 4.2. Limitations and Future Lines of Research

The limitations of this study cannot be ignored. First, the theoretical basis of coordinated development of county ecology, economy and transportation of this study should be improved in subsequent research. Second, this study lacks the data for forming a pollution

index, which is challenging to obtain, and this relevant data account for about 10% of the weight of the EI [45]. Therefore, the construction of the EI should be improved in the future. Finally, there is an inevitable crossover between the economic system and the transportation system, and the correlation and boundary between the two have not been reasonably analyzed in this study.

**Author Contributions:** This article was a collaborative effort between W.W., J.G., W.Y. and J.Z., who all contributed equally to this work. All authors have read and agreed to the published version of the manuscript.

**Funding:** This study was supported and funded by the National Natural Science Foundation of China (41701118, 41701169). The authors are grateful for the receipt of the fund.

**Institutional Review Board Statement:** Not applicable.

**Informed Consent Statement:** Not applicable.

**Data Availability Statement:** The data presented in this study are available on-demand from the first author at (wangwulin@fzu.edu.cn).

**Conflicts of Interest:** The authors declare no conflict of interest.

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
