# Peer review of "The Ecology-Economy-Transport Nexus: Evidence from Fujian Province, China"

_agriculture, doi:10.3390/agriculture12020135_

Round 1
Reviewer 1 Report
Abstract: The mention of methods and tools used (GIS based overlay analysis in this case) and analysis performed.
Line 46-50: Mention of poverty trap quote is applicable to some extent at regional scales. At local, sub national scales, saying that the economic backwardness further aggravates the ecological environment degradation, as it depends on whether it is a protected/wilderness area.
Line 86: The sentence Dog et al... Is not clear. You may rewrite to convey the context or remove it.
Line 91-93: The research relationship.... rewrite the sentence to make it clear.
Line 125: The sentence “Lucid waters and lush mountains are invaluable assets” to be deleted
Line 127: The term forest coverage rate is not clear, whether they mean forest cover change rate, pls. correct appropriately to make it explicit.
Line 147: Authors may strike off the digital maps and may rewrite as remote sensing satellite data.
Section 2.2: All three references in to be in appropriate format as per the journal guidelines.
Section 2.3.1: Number of decimal place for the indices to be made uniform, may be changed to two decimal places.
Section 2.3.2: References if any for the indicators presented need to be added if they are not developed by the authors.
Section 2.3.4: Lacks appropriate references to the concept of coupling and coordination and the equations thereof.
Section 3.1.1: a) The authors abruptly compared the Kaiser-Meyer-Olkin values of the year 2008 and 2018 without any mention of the term and its relevance in the study. To be elaborated appropriately
- b) The authors are suggested to give a separate table of the 16 indexes of EC for more clarity
- c) The authors must specify the advantage of using the PCA method while comparing the Eigen values with necessary references.
Section 2.3.4: a) This section being the major foundation of the research, need to be strengthened with proper references.
Section 3.2.1: The first sentence - According to Zhang and Wen[41]…can be moved to section 2.3.4
- b) Figure 3 is a graphical representation of the county Coordination type of the 58 counties in Fujian Province in 2000 and 2018, it should also be given in the map for making spatially explicit comparisons.
Section 4: Although it is given as conclusion, it would be better to put it under discussion and extract a separate conclusion section from it highlighting the usefulness the study as baseline r for mapping the SDG 2030 taking 2028 data as a reference. Limitations and recommendation if any based on the study should also be highlighted appropriately in this section as a way forward considering the scope of upscaling for other provinces.
Author Response
Dear Professor,
Thanks you so much for giving us an opportunity to revise this paper.
We have revised the paper carefully according the advice.
Through recombing the relevant literatures, we have modified and improved the introduction content, and the references of the paper has increased from 43 to 52.
Based on the original paper, we have improved our research design, mainly including the revision and improvement of abstract, literature review, data sources, research methods, charts, conclusions and discussions.
In order to describe the research methods accurately, We have redescribed the data resource and research methods and added relevant references.
The results are clearly presented in the revised paper, we have improved the relevant statements, charts and other content.
In section 4, we have rewrote the discussion and conclusion, including research results, contribution and limitations and future lines of research.
Best regards,
Wulin Wang
Our response (blue font) to the comments and suggestions of reviewer (black font):
Abstract: The mention of methods and tools used (GIS based overlay analysis in this case) and analysis performed.
We have modified the abstract according to expert opinion. See the modified abstract for details.
Line 46-50: Mention of poverty trap quote is applicable to some extent at regional scales. At local, sub national scales, saying that the economic backwardness further aggravates the ecological environment degradation, as it depends on whether it is a protected/wilderness area.
Agree to this suggestion. By introducing new literatures [15]~[19], we discussed the poverty trap, relationship between economic growth and ecological environment here.
Line 86: The sentence Dog et al... Is not clear. You may rewrite to convey the context or remove it.
The sentence has been revised to “Dong et al. established an integrated risk evaluation model to analyse the economic, social and ecological risks of China-Mongolia-Russia high-speed railway, determine their magnitude and spatial distribution pattern, and propose policy suggestions to reduce construction risks.”
Line 91-93: The research relationship.... rewrite the sentence to make it clear.
The sentence has been revised to “The role of ecology, economy and transportation has generated in several academic research approaches, but few studies have evaluated the coupling and coordination degree of economy, ecological environment and transportation in China from regional perspective.”
Line 125: The sentence “Lucid waters and lush mountains are invaluable assets” to be deleted
The sentence mentioned above has been deleted.
Line 127: The term forest coverage rate is not clear, whether they mean forest cover change rate, pls. correct appropriately to make it explicit.
The sentence has been revised to “Fujian Province enjoys a beautiful ecological environment where the forest coverage was 66.8% in 2021.”
Line 147: Authors may strike off the digital maps and may rewrite as remote sensing satellite data.
The sentence has been revised to “This paper analyses the ecological environment, economic and social development level and comprehensive transportation development level by using remote sensing data, statistical data, and transport infrastructure network data of Fujian Province.”
Section 2.2: All three references in to be in appropriate format as per the journal guidelines.
We have revised all references of data sources as required.
Section 2.3.1: Number of decimal place for the indices to be made uniform, may be changed to two decimal places.
We have modified the data according to the suggestion.
Section 2.3.2: References if any for the indicators presented need to be added if they are not developed by the authors.
The indicators were developed by ourselves, we have cited the Chinese reference.
Section 2.3.4: Lacks appropriate references to the concept of coupling and coordination and the equations thereof.
We have cited the related references. These number s of these references are 17, 20, 48, 49 and 50.
Section 3.1.1: a) The authors abruptly compared the Kaiser-Meyer-Olkin values of the year 2008 and 2018 without any mention of the term and its relevance in the study. To be elaborated appropriately
b) The authors are suggested to give a separate table of the 16 indexes of EC for more clarity
c) The authors must specify the advantage of using the PCA method while comparing the Eigen values with necessary references.
a) In section 3.1.1, it introduced the principal component analysis method and the function of KMO test.
b) We have made a table to describe the 16 indexes of EC in section 3.1.1.
c) We have introduced the main advantages and main steps of PCA in section 3.1.1.
Section 2.3.4: a) This section being the major foundation of the research, need to be strengthened with proper references.
a) We have combed the relevant literatures and made correct citation for strengthening the research foundation.
Section 3.2.1: The first sentence - According to Zhang and Wen[41]…can be moved to section 2.3.4
b) Figure 3 is a graphical representation of the county Coordination type of the 58 counties in Fujian Province in 2000 and 2018, it should also be given in the map for making spatially explicit comparisons.
The sentence “According to Zhang and Wen,……” has been moved to section 2.3.4.
b) Figure 3 has been changed from columnar overlapping map to type map, the type map are more conducive for making spatially explicit comparison.
Section 4: Although it is given as conclusion, it would be better to put it under discussion and extract a separate conclusion section from it highlighting the usefulness the study as baseline r for mapping the SDG 2030 taking 2028 data as a reference. Limitations and recommendation if any based on the study should also be highlighted appropriately in this section as a way forward considering the scope of upscaling for other provinces.
We agree with the expert’s suggestion. We have made significant revision in section 4. Section 4 includes the main conclusions, main contributions, limitations and future lines of research.

Reviewer 2 Report
The work introduces a good intuition with a catchy title. Three indicators of the ecological quality, the economic development and the relative regional transportation capacity are computed to model the degree of coordination at county scale in a large Fujian Province in China.
The authors venture into a risky exercise with good intentions. By and large, I liked the work but one major flaw: the way they carried out the assessment of the ecological index. One of three critical components they considered in the proposal. A linear computation among shares of land cover classes and normalised vegetation index is not enough to me to justify the ecological quality of a given piece of land.
But first thing first,
- the Abstract looks more as if written for the Results section than for the short summary of the major aspects of the entire paper in a prescribed sequence.
- lines from 48 to 52: it is a quite strong sentence and should deserve more references to be backed. The poverty-trap "per se" is an economic thought, shifting ecology wise needs to be argued in a proper way wiping away any doubt. For instance, of course, underdevelopment leads to a poor quality environment, nor income growth favours it.
- lines 78 and following: the impact of -linear- transportation infrastructures on the landscape is been widely studied as a hot topic, it should be argued more in-depth.
- lines 147 and following: use remote sensing data to compute landscape or ecology indexes to infer the ecological quality of the environment is a consolidated methodology. The authors here should be more generous in the details of the method used. What spatial resolution, what bands, what data do they analyse to "reflect the quality of the ecological environment"?
- Paragraph 2.3.1 apologies for saying it, but I do not agree at all on the overall approach and the computation approach. Moreover, there are no references upholding the way the authors have chosen to derive the comprehensive index EI.
- consequently, the paper has not made a detailed investigation (Line 378) in my opinion and (Line 417) it is not possible to conclude that the method is robust enough to bear the conclusion as at Line 419-420
That is why for the moment I have no answers on the interest to readers and on the overall merit
that's all
Author Response
Dear Professor,
Thanks you so much for giving us an opportunity to revise this paper.
We have revised the paper carefully according the advice.
Through recombing the relevant literatures, we have modified and improved the introduction content, and the references of the paper has increased from 43 to 52.
Based on the original paper, we have improved our research design, mainly including the revision and improvement of abstract, literature review, data sources, research methods, charts, conclusions and discussions.
In order to describe the research methods accurately, We have redescribed the data resource and research methods and added relevant references.
The results are clearly presented in the revised paper, we have improved the relevant statements, charts and other content.
In section 4, we have rewrote the discussion and conclusion, including research results, contribution and limitations and future lines of research.
Best regards,
Wulin Wang
Our response (blue font) to the comments and suggestions of reviewer (black font):
The work introduces a good intuition with a catchy title. Three indicators of the ecological quality, the economic development and the relative regional transportation capacity are computed to model the degree of coordination at county scale in a large Fujian Province in China.
The authors venture into a risky exercise with good intentions. By and large, I liked the work but one major flaw: the way they carried out the assessment of the ecological index. One of three critical components they considered in the proposal. A linear computation among shares of land cover classes and normalised vegetation index is not enough to me to justify the ecological quality of a given piece of land.
But first thing first,
Through literature review, we find that the research on the coupling coordination between ecology and economy, ecology and transportation, and economy and transportation is very substantial. But few studies have focused on the coupling coordination among ecology, economy and transportation. Therefore, we try to take counties of Fujian Province as an example to study the coupling coordination relationship and its evolution among the three systems in 2000 and 2018.
We constructed EI indicator to measure county ecological environment, this index is based on < Technical criterion for ecosystem status evaluation >released by Ministry of Environmental Protection of China. Due to missing data, this study lacks the data of pollution index, and this relevant data accounts for about 10% of the weight of EI. It doesn't have a fundamental effect on EI.
- the Abstract looks more as if written for the Results section than for the short summary of the major aspects of the entire paper in a prescribed sequence.
We rewrote the abstract, and the revised abstract briefly summarized the research purposes, research methods, research findings and so on in the prescribed order.
- lines from 48 to 52: it is a quite strong sentence and should deserve more references to be backed. The poverty-trap “per se” is an economic thought, shifting ecology wise needs to be argued in a proper way wiping away any doubt. For instance, of course, underdevelopment leads to a poor quality environment, nor income growth favours it.
We have collected and sorted out the literature again, and rewrote the sentences of Lines 48 to 52 as follow: The general understanding is that ecological environment degradation and economic backwardness are interrelated and geographically coupled, the coupling is presented in the form of “poverty trap”. More explicitly, the more serious the ecological environment degradation, the more backward the regional economy is, and vice versa, the economic backwardness further aggravates the ecological environment degradation[15]. However, economic growth will also cause environmental pollution. For example, China’s economic growth also pays the price of environmental pollution[16]. The relationship between economic growth and ecological environment improvement can not be generalized. In different cities or regions, there is not only a harmonious relationship between advanced economy and good ecological environment, but also a low-level coupling and coordinated state, and the coordination between economy and environmental ecology is the necessary condition for sustainable development[17,18]. The high production and living functions, and the low ecology function mainly occurred in well-developed regions, as well as the strong trade-offs between the above two functions and ecology function[19].
- lines 78 and following: the impact of -linear- transportation infrastructures on the landscape is been widely studied as a hot topic, it should be argued more in-depth.
We have argued and rewrote the sentences of Lines 78. The cited literatures increased from 6 to 10, and the length of the paragraph increased from 13 lines to 21 lines. See the revised papers for details.
- lines 147 and following: use remote sensing data to compute landscape or ecology indexes to infer the ecological quality of the environment is a consolidated methodology. The authors here should be more generous in the details of the method used. What spatial resolution, what bands, what data do they analyse to “reflect the quality of the ecological environment”?
The revised paper introduces the data sources in more detail and accurately. In section 2.2. it has described the data band, spatial resolution, etc.
- Paragraph 2.3.1 apologies for saying it, but I do not agree at all on the overall approach and the computation approach. Moreover, there are no references upholding the way the authors have chosen to derive the comprehensive index EI.
The indicator of EI is constructed by the <Technical Criterion for Ecosystem Status Evaluation> issued by the Ministry of Environmental Protection of China in 2015. We have quoted the criterion correctly. In our paper, we have lost pollution data which account for 10% weight of EI, it has little influence on the evaluation of EI.
- consequently, the paper has not made a detailed investigation (Line 378) in my opinion and (Line 417) it is not possible to conclude that the method is robust enough to bear the conclusion as at Line 419-420
We may have used the wrong words, and we have made corresponding revisions. We have deleted the sentence in Line 378 of the original paper. The sentences of Line 417 of original paper are revised as follows: “Fisrt, The overall conclusions are that the parameters used and calculation method developed in this study provide sufficiently robust indicators to track the sustainability of ecological environment, …”
That is why for the moment I have no answers on the interest to readers and on the overall merit.
that’s all
We revised the paper as best we could. Thank you very much for the your insights.

Round 2
Reviewer 1 Report
I feel that authors have taken care of the comments given earlier
thus, the MS can be considered for final publications after the
minor English corrections (if needed) at your end.
Author Response
Thanks to the reviewer‘s detailed comments. According to the reviewer's comments, we have revised the paper carefully.
Reviewer 2 Report
Thanks to the authors for considering my comments and revising the entire manuscript.
I'm afraid I still have to disagree with how the authors compute the ecological quality. On the other hand, to point out further on that matter might lead us into an endless discussion. I would instead prefer to lay off the issue. I am therefore satisfied with the new way the authors have put it in the general context.
my last comments are as follow:
- line 154: If I were the authors, I would avoid sentences like 'beautiful ecological environment': the beautiful scenery of a place little has to do with its ecological functionality
- line 171: could the authors please explain the nature of "no data" in figure 1. I might be mistaken but I have found no references.
that's all
Author Response
Response to Reviewer 2 Comments
Thanks to the authors for considering my comments and revising the entire manuscript. I'm afraid I still have to disagree with how the authors compute the ecological quality. On the other hand, to point out further on that matter might lead us into an endless discussion. I would instead prefer to lay off the issue. I am therefore satisfied with the new way the authors have put it in the general context.
Thanks for the reviewer’s professional guidance. We have considered all aspects of the reviewer’s opinion, and revised the paper as much as possible. Although there are still some shortcomings in the paper, we will carefully verify it in future research.
my last comments are as follow:
- line 154: If I were the authors, I would avoid sentences like 'beautiful ecological environment': the beautiful scenery of a place little has to do with its ecological functionality
“beautiful ecological environment” is a customary or commonly used Chinese expression, that is “优美的生态环境”. To avoid ambiguity, we deleted the word “beautiful” in this sentence.
- line 171: could the authors please explain the nature of "no data" in figure 1. I might be mistaken but I have found no references.
that's all
We added the following contents to the paper to explain “no data” of Figure 1: In Figure 1, the “No data” areas mainly refer to the municipal districts of the 9 prefecture-level cities and Jinmen county.
